# Interventions to improve awareness and reduce the stigma associated with neurodegenerative conditions in minority ethnic communities: A scoping review protocol

Maria Caulfield[1,2,3*◉], Sahdia Parveen[1,2,3‡], Matthew Prina[1,4‡], Jan R Oyebode[1,2,3‡], Karen Windle[1,5], Catherine Charlwood[1,6,7‡], Adelina Comas-Herrera[1,8], Catherine Quinn[1,2,3], Linda Clare[1,6,7]

1 NIHR Policy Research Unit in Dementia and Neurodegeneration University of Exeter (DeNPRU Exeter), United Kingdom, 2 Centre for Applied Dementia Studies, University of Bradford, United Kingdom, 3 Wolfson Centre for Applied Health Research, Bradford, United Kingdom, 4 Population Health Sciences Institute, Faculty of Medical Sciences, Newcastle University, Newcastle Upon Tyne, United Kingdom, 5 School of Nursing and Paramedic Science, Faculty of Health and Life Sciences, University of Ulster, United Kingdom, 6 Department of Health and Community Sciences, University of Exeter Medical School, Exeter, United Kingdom, 7 NIHR Applied Research Collaboration South-West Peninsula, United Kingdom, 8 Care Policy and Evaluation Centre, London School of Economics and Political Science, United Kingdom

◉ These authors contributed equally to this work.
‡ MP, JRO and CC also contributed equally to this work.
* m.c.caulfield@bradford.ac.uk

## Abstract

### Objective

This scoping review aims to identify interventions aiming to improve awareness of and reduce stigma related to neurodegenerative conditions within South Asian and Black (African-Caribbean, African, African American, Black British) communities with a focus on synthesising the methods employed for culturally tailoring interventions.

### Introduction

Minority ethnic communities affected by neurodegenerative conditions often face health and social care disparities. This can lead to delayed diagnosis and poor health outcomes. Interventions that provide relevant, accessible information about neurodegenerative conditions may help reduce disparities in care access. There is limited knowledge about the methods used to culturally tailor interventions for minority ethnic communities and their efficacy.

### Inclusion criteria

Eligible sources will include interventions specifically tailored for South Asian and Black communities, living with dementia, Parkinson's disease, Huntington's disease, or motor neurone disease. Interventions must be conducted in countries that

**Data availability statement:** No datasets were generated or analysed during the current study. All relevant data from this study will be made available upon study completion.

**Funding:** This research is funded through the NIHR Policy Research Unit in Dementia and Neurodegeneration University of Exeter, reference NIHR206120. The views expressed are those of the authors and not necessarily those of the NIHR or the Department of Health and Social Care. The funders had no role in study design, data collection and analysis, decision to publish, or preparation of the manuscript.

**Competing interests:** The authors have declared that no competing interests exist.

are member states of the Organisation for Economic Co-operation and Development where these two groups constitute minority populations and are likely to face inequalities in care access.

## Methods

A scoping review guided by the Joanna Briggs Institute Manual for Evidence Synthesis will be conducted. Searches of Medline (EBSCO), APA PsycInfo (EBSCO), and EMBASE (Elsevier) will be conducted. Study selection will be based on 100% agreement between two reviewers. Data will be extracted, charted, and summarised narratively followed by consultation with stakeholders.

## Implications

This review will identify culturally sensitive strategies for raising awareness and reducing the stigma associated with neurodegenerative conditions among South Asian and Black communities within the Organisation for Economic Co-operation and Development countries. By utilising these inclusive approaches, communities may feel more empowered to seek a diagnosis for symptoms and live better with the condition. The findings of this review will be shared with the public and policymakers to promote awareness and evidence-based policy making.

## Introduction

Neurodegenerative conditions are a public health concern affecting millions of people worldwide [1,2]. These conditions are characterised by the progressive degeneration of the structure and function of the nervous system, leading to a decline in cognitive and physical function. Globally, the most prevalent condition is dementia, which affects an estimated 55 million people [3]. Alzheimer's disease is the primary cause of dementia among adults aged 65 years and older, accounting for approximately 60% to 70% of cases [3]. Other prevalent forms of dementia include vascular dementia, dementia with Lewy bodies, and mixed dementia, while rare dementias, including Creutzfeldt-Jakob disease and frontotemporal dementia, proportionally affect more adults under the age of 65 years. Parkinson's disease, the second most common neurodegenerative disease, has shown the fastest increase in prevalence and disability [4], and affects approximately 10 million people [5]. Globally, there are approximately 268,673 cases of motor neurone disease [6], and Huntington's disease affects 3.92 individuals per 100,000 worldwide [7]. As the global population ages, the prevalence of these conditions is likely to increase [8–10].

While these conditions affect individuals of all ethnicities, disparities in timely diagnosis and access to health and social care are experienced by minority ethnic communities living with these conditions and contribute to poor health outcomes and quality of life [11–15]. A minority ethnic group is a group of people who share a common cultural identity, traditions, and languages, distinguishing them from the

majority ethnic group in a particular country [16]. Addressing disparities across the whole care pathway requires a multi-faceted response. A critical first step in this effort is to facilitate greater awareness of conditions and their symptoms and address cultural stigmas that can undermine help seeking behaviours [17–19]. Awareness interventions could be a valuable mechanism to achieve these goals [20–22]. However, the ways in which interventions are tailored for minority ethnic communities to reflect their cultural values, practices, or understanding of disease aetiology and their symptoms, as well as intervention effectiveness, is poorly understood. This lack of knowledge exchange hampers the adaptation and scaling up of effective methods for underserved groups.

The population of the United Kingdom (UK) is composed of a wide range of ethnic groups. Around 18% of the UK population are from a minority ethnic background, with South Asian and Black ethnicities being the two largest minority groups [23]. Estimates for the UK indicates around 25,000 individuals from minority ethnic communities living with dementia, projected to rise to over 50,000 by 2026 and 172,000 by 2051 [24]. More recently, using data from electronic health records, Mukadam et al. [25] report that dementia incidence is higher among Black people compared to white people and Black and South Asian people living with dementia have a younger age of death than white people. Approximately 153,000 people in the UK have Parkinson's disease, with up to 1 in 20 of Black, Asian, or mixed heritage [26]. The proportion of people from a minority ethnic background living with Huntington's disease or motor neurone disease in the UK is not known. Overall, UK population estimates for Huntington's disease and motor neurone disease are around 8,000 [27] and 5,000 [28], respectively. The analysis of incidence of motor neurone disease in England between 1998 and 2019 by Burchardt et al. [10] found that age-standardised incidence was similar in Bangladeshi, Black Caribbean, Indian, Pakistani, other Asian, and white people, but lower in Black African and Chinese people. The findings suggest that the incidence of motor neurone disease in England has increased over time, which may be due to better case detection and an ageing population.

UK policy places emphasis on early diagnosis of neurodegenerative diseases and early interventions to mitigate the impact of these conditions on individuals, families, and health and social care systems [29–31]. Culturally tailored interventions are an essential component of promoting equitable and timely access to diagnosis and support for minority ethnic groups [32,33].

In this scoping review protocol, an intervention is defined as any resource or activity designed to enhance awareness and understanding of a neurodegenerative condition. Examples of interventions may include educational lectures, informational leaflets, videos, public health campaigns, or community events. Culturally tailored interventions are those that align the content and/or the delivery of messages with the values, beliefs, and practices of a specific ethnic group. The content may incorporate faith-based considerations or use the community's language, while the intervention's setting is made accessible and relevant to the target community. For example, Zheng et al. [34] produced a culturally tailored short film about dementia, designed for Chinese American participants. The film, produced in Cantonese, centred on an elderly Chinese couple, and was screened at a local church.

The objective of this scoping review is to identify interventions to raise awareness of and reduce the stigma associated with neurodegenerative conditions developed specifically for South Asian and Black communities in the 38 member countries of the Organisation for Economic Co-operation and Development (OECD). Akin to the UK, these ethnic groups also constitute minoritised communities in the OECD member countries, making up less than 30% of the overall population. Their varied cultural practices, traditions, and languages distinguish them from the dominant societal context, and they are therefore likely to have reduced awareness of how to access healthcare.

A preliminary search of the published literature was undertaken on 06.02.2024 in Google Scholar, PROSPERO, the Cochrane Database of Systematic Reviews, and Joanna Briggs Institute (JBI) Evidence Synthesis using the key words neurodegenerative disease, minority ethnic, and intervention. No current scoping reviews or reviews in progress on the topic were identified. One systematic review, conducted by Huggins et al. [20] was identified; this assessed whether existing educational interventions increase knowledge about dementia among minority ethnic communities. Most of the studies were conducted in the USA, primarily concentrating on Black and Hispanic/Latino communities. The three studies

conducted in the UK, targeted Black (Black British, Black African, Black Caribbean, African and Caribbean) and Asian (British Indian, Asian Caribbean) populations. The interventions predominantly took a psychoeducational approach, with the aims of enhancing knowledge regarding cognitive assessments, fostering an understanding of person-centred dementia care, and teaching coping strategies to families and unpaid carers of people living with dementia to reduce burden and stress. Only two studies, Parveen et al. [35] and Roche et al. [36], specifically addressed the effectiveness of interventions aimed at enhancing knowledge of dementia in South Asian families and promoting timely help-seeking for dementia in older Black African and Caribbean adults, respectively. This systematic review was restricted to studies published between 2015 and 2020, focusing solely on knowledge about dementia. To broaden the scope and to expand the period covered by the searches, the present scoping review will consider four neurodegenerative conditions, specifically dementia of any type, Parkinson's disease, Huntington's disease, and motor neurone disease, and consider studies published from inception to 2025.

### Review questions

The review will address three questions:

1. What interventions have been developed to improve awareness of and/or reduce stigma associated with neurodegenerative conditions within South Asian and Black (African-Caribbean, African, African American, Black British) communities in OECD countries?

2. What cultural adaptations were undertaken, considering factors such as setting, content, mode of delivery?

3. What evidence is there for the feasibility or effectiveness of the interventions?

## Method

The proposed scoping review will be conducted in accordance with the JBI methodology for scoping reviews [37] and be prepared in accordance with the PRISMA scoping review checklist (S4 File) [38].

### Eligibility criteria

The inclusion and exclusion criteria are presented in Table 1.

### Search strategy

Relevant studies will be identified by searching three electronic databases: Medline (EBSCO), APA PsycInfo (EBSCO), and EMBASE (Elsevier). No date limitations will be applied. An initial limited search in Medline (EBSCO) was undertaken to identify studies on interventions for South Asian and Black communities. The text words found in the titles and abstracts of pertinent articles, as well as the index terms chosen to describe the articles, were used to construct a full search strategy (S1 File). A subject librarian, MC and SP developed a search strategy for the concepts: minority ethnic communities, neurodegenerative conditions, intervention. The search strategy was piloted in Medline (EBSCO), APA PsycInfo (EBSCO), and EMBASE (Elsevier) and based on this initial search, search terms were adjusted and refined as necessary. Forward citation screening of included full-text studies will be conducted. The screening of the published and grey literature will be undertaken by two researchers (MC and SP).

We expect that many interventions will originate from local or regional initiatives, which may not be published in peer-reviewed academic journals. We will conduct a comprehensive and systematic search of grey literature to identify non-peer-reviewed sources such as reports, newsletters, case studies, leaflets, video, and webinars etc. The grey literature search will have two components: an advanced Google search of a preselected list of 77 websites and a broader Google web search.

**Table 1. Inclusion and exclusion criteria.**

| Criteria | Inclusion | Exclusion |
|---|---|---|
| Population 1 | Interventions designed for people of any age or gender from a South Asian or Black ethnicity.<br>People of South Asian descent originating from Afghanistan, Bangladesh, Bhutan, India, Maldives, Nepal, Pakistan, and Sri Lanka.<br>Those regarded as Black include those identifying as Black British, African American, Black Caribbean, or Black African. | Interventions that target health or social care professionals. |
| Population 2 | Interventions that focus on dementias of any type, Parkinson's disease, Huntington's disease, or motor neurone disease. | |
| Intervention | Interventions that take the form of an educational programme, health promotion, leaflet, campaign, community workshops, roadshows, or other means of creating awareness or reducing stigma of the condition within the public. | Interventions that involve the delivery of treatments or therapies for individuals and/or families affected by neurodegenerative conditions. |
| Context | Interventions delivered within the member countries of the OECD: Australia, Austria, Belgium, Canada, Chile, Colombia, Costa Rica, Czech Republic, Denmark, Estonia, Finland, France, Germany, Greece, Hungary, Iceland, Ireland, Israel, Italy, Japan, South Korea, Latvia, Lithuania, Luxembourg, Mexico, Netherlands, New Zealand, Norway, Poland, Portugal, Slovak Republic, Slovenia, Spain, Sweden, Switzerland, Turkey, United Kingdom, United States. | |
| Types of sources | Quantitative design,<br>Qualitative design,<br>Mixed methods design,<br>Systematic or scoping reviews,<br>Grey literature sources. | Opinion papers,<br>Proceedings,<br>Protocols,<br>Published conference abstracts. |

The 77 websites are either the main English language condition-specific websites for dementia, Parkinson's disease, Huntington's disease, and motor neurone disease (e.g., Alzheimer Europe, International Huntington's Disease Association), hosted by organisations in the UK, the United States, Canada, Europe, and Australia, or non-condition specific websites for global health organisations and health and social care think tanks (e.g., Kings Fund, Health Foundation). The list of websites can be found in supplementary material (S2 File). The search of the 77 websites will use advanced Google search parameters and will be restricted to PDF documents only. An example search string is: site:alzheimers.org.uk "minority ethnic" OR "ethnic minority" OR BAME OR Black OR "South Asian" filetype:pdf. As the most applicable PDFs will be returned at the top of the search engine results page, the first 20 PDFs that Google returns for each website will be downloaded and screened.

A follow-up Google search will be conducted to find information not available in PDF format but accessible on organisation websites. This includes resources like videos, podcasts, webinars, and descriptions of interventions. The search will pair the four conditions—dementia, Parkinson's disease, Huntington's disease, and motor neurone disease—with the terms 'Black', 'South Asian', or 'Minority ethnic' combined with 'Stigma' or 'Awareness'. The first 30 search results for each combination will be reviewed.

## Evidence selection

The search results and selection process will be reported in full and presented in a Preferred Reporting Items for Systematic Reviews and Meta-analyses extension for scoping review (PRISMA-ScR) flow diagram [38]. All study references will be exported to the software Covidence, and duplicates will be removed. The Covidence software will support the management of the screening process for both titles and abstracts, as well as full texts. Two reviewers will independently screen all article titles and abstracts to identify potentially relevant articles. These articles will then undergo full text screening by the same two reviewers, who will assess them against the eligibility criteria. Disagreement between the reviewers during

the screening and selection process will be resolved through discussion or by involving an additional reviewer. Study selection will be based on consensus between the reviewers.

For grey literature searches, PDFs will be downloaded into folders, one folder for each website searched. Details of each search will be recorded in an Excel spreadsheet that includes the search date, the name of the researcher who conducted the search, the search string(s) used for each website, the number of PDFs returned in the search engine results page, the number of duplicates manually removed, and the number of PDFs extracted for screening. One reviewer will conduct the initial screening.Sources identified as potentially relevant will then undergo full-text screening by the same reviewer, who will discuss their findings with a second reviewer to ensure consistency. In cases where inclusion is unclear, eligibility decisions will be reached through group discussion involving a third reviewer.

## Data extraction

One reviewer will undertake data extraction using a form guided by JBI recommendations [37]. This form will record intervention characteristics, including where applicable, authorship, publication year, year of intervention delivery, country of origin, target population, aim(s), and methods.

Additional detail relevant to the review questions will be extracted, encompassing:

• Content of intervention: the nature of information provided.

• Mode of delivery: online, in person, or through distribution of materials.

• Setting of intervention delivery.

• Duration of delivery: time limited or continuous.

• Reach: does the intervention operate at the individual, interpersonal, community, or societal level.

• Outcomes: measured outcomes and qualitative data on the experience of developing and delivering the intervention.

• Cultural adaptations.

The extraction form will undergo a pilot phase, and any necessary modifications will be made and reported within the scoping review.

## Risk of bias assessment

In line with JBI guidelines for conducting scoping reviews, no risk of bias assessment will be conducted.

## Data analysis and presentation

The findings will be displayed in tabular format, which will aid in visually representing and outlining the methods employed for culturally adapting interventions. Additionally, a descriptive narrative will be provided to address the three review questions.

## Stakeholder consultation

Central to the methodology of scoping reviews is stakeholder consultation [39,40]. In line with this, four stakeholder workshops will be conducted via online platforms such as Zoom or Microsoft Teams. The workshops will involve the following groups:

Group 1. People affected by neurodegenerative conditions of African-Caribbean or African heritage.

Group 2. People affected by neurodegenerative conditions from South Asian backgrounds.

Group 3. People working for local community groups or as regional representatives of national organisations, supporting minority communities, or possessing specialist knowledge of neurodegenerative conditions.

Group 4. Representatives from national level organisations with specialist expertise in neurodegenerative conditions, those with public health or health communication expertise and policy makers.

The consultation workshops will follow the outline presented in Fig 1:

Each session is expected to last approximately two hours (including breaks). Participants will be asked to consent to the recording of the discussion, which will only be used for research and reporting purposes. Any quotes used will be anonymised, and the recording will not be shared publicly.

## Patient and Public Involvement

A key element of this review is the involvement of people with lived experience of neurodegenerative conditions, known as 'experts by experience'. This project is part of the wider work of the NIHR (National Institute of Health Research) Policy Research Unit in Dementia and Neurodegeneration University of Exeter (DeNPRU Exeter). DeNPRU Exeter is guided by an active group of experts by experience of which two members are attached to this project. One member is of South Asian origin (Pakistani), living with young-onset Parkinson' Disease and the second is also of South Asian origin (Indian), a former carer for her mother who lived with dementia. We are actively engaging with African-Caribbean and African communities to involve further experts by experience in this project. The role of the experts by experience will be to co-facilitate workshops with stakeholders and develop outputs for UK policymakers.

## Equality impact assessment

An equality impact assessment (EIA) is an evidence-based approach designed to help organisations ensure that their policies, practices, events, and decision-making processes are fair, do not present barriers to participation and do not disadvantage any protected groups from participation [41]. Following guidance set out by NIHR and UKRI (UK Research and Innovation), a thorough EIA has been undertaken to consider if this research will have a positive or negative impact on

**Introduction of identified interventions**

- A summary of promising interventions from the scoping review on raising awareness of neurodegenerative conditions will be presented.

**Participant insights and perspectives**

- Participants will be invited to share their insights on improving the interventions, including preferred delivery methods, expected outcomes, and other relevant factors based on their experience.

**Intervention implementation**

- Participants will discuss how to implement these interventions and identify potential barriers and facilitators to their delivery.

**Fig 1. Potential outline for consultation workshops.**

protected communities. The assessment involves considering existing evidence and consultation with protected groups. We considered the impact of this research on the following protected groups: age, disability, gender, race and ethnicity, marriage and civil partnership, sexual orientation, socio-economic status, maternity/paternity, and caregiving responsibilities. Following consideration of impact, strategies were designed to reduce negative impact on the protected group. Examples include:

- Ethnicity: It was noted that published literature often reported an individual's religion as opposed to ethnicity. Religion does not indicate an individual's ethnicity as religious groups include a wide range of ethnicities and cultures. For this review, actions to address impact will include checking the ethnicity of the participants, where reported, and not be misled by the religion reported. For the workshops, it is crucial to consider religious holidays and times of religious observance. For example, we would avoid scheduling workshops on Friday afternoons due to the prayer times for Muslims.

- Socio economic status: As with minority ethnic communities, those from disadvantaged economic groups experience barriers in participating in research. On balance, virtual workshops may offer the most practical approach in terms of time, resources, and reach for such groups. However, if needed, one-on-one, in-person discussions can be arranged for people who wish to participate but are unable to join online.

- Caregiving responsibilities: The unpredictable nature of caregiving means that flexibility is required as last-minute changes are common. After consulting with some unpaid family carers, we determined that a suitable window is between 1pm and 3pm. However, for those who are still employed, early evening sessions might be more convenient. To ensure all unpaid carers can participate, it will be important to provide alternative options, such as telephone calls, for those unable to attend in person.

  The EIA (S3 File) will be published on the University of Bradford website.

## Supporting information

**S1 File. Search strategy.**
(DOCX)

**S2 File. List of 77 websites.**
(DOCX)

**S3 File. Equality impact assessment.**
(PDF)

**S4 File. PRISMA-P 2015 checklist.**
(DOCX)

## Acknowledgments

The authors acknowledge the contribution of Dr. Anthony Martyr, Senior Research Fellow, University of Exeter Medical School, in the development of the systematic grey literature search strategy.

## Author contributions

**Conceptualization:** Sahdia Parveen.

**Funding acquisition:** Jan R Oyebode, Linda Clare.

**Writing – original draft:** Maria Caulfield.

**Writing – review & editing:** Sahdia Parveen, Matthew Prina, Jan R Oyebode, Karen Windle, Catherine Charlwood, Adelina Comas-Herrera, Catherine Quinn, Linda Clare.

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
