## [Decision Letter · Decision Letter 0]

2 Jan 2025

PONE-D-24-42224Interventions to improve awareness and reduce the stigma associated with neurodegenerative conditions in minority ethnic communities: A scoping review protocol.PLOS ONE

Dear Dr. Caulfield,

Thank you for submitting your manuscript to PLOS ONE. After careful consideration, we feel that it has merit but does not fully meet PLOS ONE’s publication criteria as it currently stands. Therefore, we invite you to submit a revised version of the manuscript that addresses the points raised during the review process.

We look forward to receiving your revised manuscript.

Kind regards,

Vincenzo De Luca

Academic Editor

PLOS ONE

2. Thank you for stating the following financial disclosure:  [This research is funded through the NIHR Policy Research Unit in Dementia and Neurodegeneration University of Exeter, reference NIHR206120. The views expressed are those of the authors and not necessarily those of the NIHR or the Department of Health and Social Care.]. 

Additional Editor Comments (if provided):

Reviewers' comments:

Reviewer's Responses to Questions

**Comments to the Author**

1. Does the manuscript provide a valid rationale for the proposed study, with clearly identified and justified research questions?

Reviewer #1: Yes

2. Is the protocol technically sound and planned in a manner that will lead to a meaningful outcome and allow testing the stated hypotheses?

Reviewer #1: No

3. Is the methodology feasible and described in sufficient detail to allow the work to be replicable?

Reviewer #1: No

4. Have the authors described where all data underlying the findings will be made available when the study is complete?

Reviewer #1: Yes

5. Is the manuscript presented in an intelligible fashion and written in standard English?

Reviewer #1: Yes

6. Review Comments to the Author

You may also provide optional suggestions and comments to authors that they might find helpful in planning their study.

Reviewer #1: The manuscript proposes a scoping review that is used to evaluate interventions to increase awareness and reduce stigma related to neurodegenerative conditions in minority groups. These disparities faced in these groups often lead to delayed diagnosis and poor health outcomes.

However, there're some severe issues I can see in the manuscript:

1. the definition of "culturally tailored interventions" is lacking in the manuscript. A more detailed definition should be provided to help readers understand what the authors are talking about. Or the authors can provide some examples of such interventions. If the authors can provide some quantitative criteria to define "culturally tailored interventions", that would be even better.

2. the inclusion of grey literature is unclear. Google search can be biased. A better, more objective criterion should be set to address this issue.

3. the inclusion criteria of stakeholder input might be too limited. I understand that policy makers, practitioners and the minority people should be included. But representatives from the major public should also be included, as the intervention will inevitably show up in public and it's a must to consider if the intervention can also be not just accepted by the major public and even spread/shared by them to other people. These types of communication will accelerate/improve the impact of such interventions in the whole society.

7. PLOS authors have the option to publish the peer review history of their article (what does this mean? ). If published, this will include your full peer review and any attached files.

**Do you want your identity to be public for this peer review?** For information about this choice, including consent withdrawal, please see our Privacy Policy .

Reviewer #1: No

---

## [Author Response · Author response to Decision Letter 1]

10 Jan 2025

1.The definition of "culturally tailored interventions" is lacking in the manuscript. A more detailed definition should be provided to help readers understand what the authors are talking about. Or the authors can provide some examples of such interventions. If the authors can provide some quantitative criteria to define "culturally tailored interventions", that would be even better.

Author response: On page 6, line 117, we have expanded our definition of an ‘intervention’ and provided an example to illustrate what a culturally tailored intervention might entail.

2.The inclusion of grey literature is unclear. Google search can be biased. A better, more objective criterion should be set to address this issue. On page 9, line 188, we have outlined our approach to searching the grey literature as explicitly as possible.

Author response: We believe our method is comprehensive and thorough compared to many other scoping reviews, which often only state that a search of the grey literature was conducted without providing further detail. We will adopt a two-pronged approach.

First, we will use advanced Google search parameters on a preselected list of 77 websites. The list of websites and an example search string are provided. Since the most relevant PDFs are typically prioritised in Google search results, we will download and screen the first 20 PDFs returned for each website.

Second, we will conduct a broader Google web search. We have specified the key terms and pairings we will use, as well as the number of search results we will review.

Together, these methods will help identify a diverse range of grey literature sources, including PDFs, videos, podcasts, webinars, and other online resources.

3.The inclusion criteria of stakeholder input might be too limited. I understand that policy makers, practitioners and the minority people should be included. But representatives from the major public should also be included, as the intervention will inevitably show up in public and it's a must to consider if the intervention can also be not just accepted by the major public and even spread/shared by them to other people. These types of communication will accelerate/improve the impact of such interventions in the whole society.

Author response: As this is the first workstream of a policy research project, our stakeholder focus is experts-by-experience who have lived experience of neurodegenerative conditions and those already working with or for ethnic minority communities affected by neurodegenerative conditions. Experts-by-experience who have already been affected by awareness campaigns, for better or worse, are in a strong position to advise on what works. Many of them have been actively involved in raising awareness within their own communities since being affected by neurodegeneration and so again have important perspectives to share. Given that experts-by-experience represent different conditions, we seek to uncover ways in which awareness raising is working for certain conditions and where best practice might be adopted for others.

Local/regional practitioners and national Equality, Diversity, and Inclusion (EDI) leads (or those with EDI as part of their portfolio) will be consulted as they have a real-time view of what is and is not working on the ground and the scale of the problem and can share insights into efforts which have yielded positive results, and wrong turns taken. UK national leads for EDI are in positions to create policy changes for their organisations and influence public perceptions.

As the focus of this project was never to identify and promote one single awareness intervention, the broader public will not be consulted in this instance. However, the authors would strongly recommend that any original research investigating the feasibility of implementing particular interventions include this group in their stakeholder consultations.

---

## [Decision Letter · Decision Letter 1]

17 Mar 2025

Interventions to improve awareness and reduce the stigma associated with neurodegenerative conditions in minority ethnic communities: A scoping review protocol.

PONE-D-24-42224R1

Dear Dr. Caulfield,

We’re pleased to inform you that your manuscript has been judged scientifically suitable for publication and will be formally accepted for publication once it meets all outstanding technical requirements.

Kind regards,

Vincenzo De Luca

Academic Editor

PLOS ONE

Additional Editor Comments (optional):

Reviewers' comments:

Reviewer's Responses to Questions

**Comments to the Author**

1. Does the manuscript provide a valid rationale for the proposed study, with clearly identified and justified research questions?

Reviewer #1: Yes

2. Is the protocol technically sound and planned in a manner that will lead to a meaningful outcome and allow testing the stated hypotheses?

Reviewer #1: Yes

3. Is the methodology feasible and described in sufficient detail to allow the work to be replicable?

Reviewer #1: Yes

4. Have the authors described where all data underlying the findings will be made available when the study is complete?

Reviewer #1: Yes

5. Is the manuscript presented in an intelligible fashion and written in standard English?

Reviewer #1: Yes

6. Review Comments to the Author

You may also provide optional suggestions and comments to authors that they might find helpful in planning their study.

Reviewer #1: All concerns have been adequately addressed, and I am happy to recommend this manuscript for acceptance.

7. PLOS authors have the option to publish the peer review history of their article (what does this mean? ). If published, this will include your full peer review and any attached files.

**Do you want your identity to be public for this peer review?** For information about this choice, including consent withdrawal, please see our Privacy Policy .

Reviewer #1: No

---

## [Editor Report · Acceptance letter]

PONE-D-24-42224R1

PLOS ONE

Dear Dr. Caulfield,

I'm pleased to inform you that your manuscript has been deemed suitable for publication in PLOS ONE. Congratulations! Your manuscript is now being handed over to our production team.

Kind regards,

on behalf of

Dr. Vincenzo De Luca

Academic Editor

PLOS ONE